

# Palmprint recognition based on principal line features

Hongxia Wang and Teng Lv

School of Big Data and Artificial Intelligence, Anhui Xinhua University, Hefei, Anhui, China

## ABSTRACT

With the increasing prevalence and diversity of imaging devices, palmprint recognition has emerged as a technology that better meets the demands of the modern era. However, traditional manual methods have limitations in effectively extracting palmprint principal line features. To address this, we introduce a novel data augmentation method. First, the wide line extraction (WLE) filter is utilized to specifically target and extract the prominent principal lines of palmprints by leveraging their direction and width characteristics. Then, a Gabor filter is applied to the WLE-extracted results to purify the features and remove fine lines, as fine lines can introduce noise and redundancy that interfere with the accurate extraction of significant principal line features crucial for palmprint recognition. Evaluating this data augmentation across four common Vision Transformer (ViT) classification models, experimental results show that it improves the recognition rates of all databases to varying degrees, with a remarkable 32.9% increase on the high-resolution XINHUA database. With the successful removal of fine lines by WLE, we propose a new Layer Visual Transformer (LViT) design paradigm. For its input, distinct blocking strategies are adopted, carefully designed to partition the data to capture different levels of spatial and feature information, using larger blocks for global structure and smaller ones for local details. The output results of these different blocking strategies are fused by "sum fusion" and "maximum fusion", and the local and global features are effectively utilized by combining complementary information to improve the recognition performance and get state-of-the-art results on multiple databases. Moreover, LViT requires fewer training iterations due to the synergistic effects of the blocking strategies, optimizing the learning process. Finally, by simulating real-world noise conditions, we comprehensively evaluate LViT and find that, compared with traditional methods, our approach exhibits excellent noise-resistant generalization ability, maintaining stable performance across the PolyU II, IIT Delhi, XINHUA, and NTU-CP-V1 databases.

## INTRODUCTION

Palmprint recognition is a critical research topic in the field of biometrics. It involves collecting palmprint images, extracting their features, as well as comparing and matching these features to achieve to achieve identity authentication. Palmprint recognition has important application value in the fields of finance, government, medical care, education

Corresponding author
Hongxia Wang,
wanghongxia@axhu.edu.cn

and other domains (*Minaee et al., 2023*). For instance, palmprint recognition can be employed in banks for identity verification, in governments for citizen authentication, in hospitals to enhance patient privacy protection, and in schools to secure student records, among other applications. Compared with other biometric technologies, palmprint has a more stable shape and unique texture information, and offers advantages including easy collection, low cost, and high security. According to whether the palm touches the equipment or not, the palmprint acquisition can be divided into contact and non-contact methods. Compared with non-contact acquisition, contact acquisition helps to ensure the stability of the sensor and the acquisition environment, resulting in better image quality. With the popularization and diversification of imaging equipment, non-contact acquisition can avoid contact between the palm and any object, resulting in low invasiveness, high reliability, and strong social acceptance. At the same time, it makes the palmprint image acquisition more flexible and convenient, and can also provide sufficient palm print information, so it has gradually become the mainstream approach in palmprint recognition systems (*Fei et al., 2019*; *Li et al., 2024*).

As we all know, palmprints are composed of wrinkles and principal lines. The principal lines can be used as an independent feature of the palm. Therefore, there are several reasons to carefully study the method based on the principal lines. Firstly, the method based on the principal lines can be integrated with human behavior. For example, when humans compare two palmprints, they visually compare the principal lines. Secondly, principal lines are generally more stable than wrinkles. Wrinkles are easily concealed by poor lighting conditions, compression artifacts, and noise in real-world environments. Thirdly, the principal line can be used as an important component of multi-feature methods. Finally, due to their simplicity, principal lines can be used for palmprint classification or fast retrieval systems. However, up to now, the method based on the principal line has not been fully studied. The main reason is that complex palmprint images often contain prominent and extensive wrinkles, which makes it challenging to extract the principal lines. Additionally, many researchers believe that it is difficult to achieve high recognition rates only by using the principal line due to the similarity among individuals (*Zhang et al., 2003*). However, there have been no relevant experiments conducted to verify their views.

Obviously, lines are the basic features of palmprints. Therefore, line-based methods also play a significant role in the field of palmprint verification and recognition. Due to slight variations in pose, rotation angle, and illumination intensity during non-contact image acquisition, directly using the extracted principal lines for matching verification often yields unsatisfactory results. The common matching method is to measure the similarity distance between the principal lines of two samples. Because the process of extracting the principal lines is rough, the extracted principal lines are often purified first, but there is still no unified method to effectively deal with all kinds of fine lines, so it is urgent to find a universal matching scheme. Deep learning is a good way to learn target features from a local perspective, which is usually not affected by slight deformations, distortion, translation and other factors. Therefore, the principal line discrimination based on deep learning can meet the needs of the palm print recognition system with non-contact and

union constraints better than traditional methods. As a paradigm shift in the field of computer vision, Vision Transformers (ViTs) use the transformer architecture to achieve first-class performance in image classification tasks, while providing efficiency, flexibility, interpretability and scalability. This article applies ViTs to palmprint principal line matching tasks, and achieves state-of-the-art results in three databases.

This article mainly studies deep learning palmprint recognition algorithm. The main contributions of this research are as follows.

(1) We propose a new filter named wide line extraction (WLE). This innovation is of great significance, because before our work, the existing palmprint line extraction methods failed to fully consider the width and direction characteristics of palmprint lines. WLE filtering can extract more prominent lines from palmprint images. Specifically, WLE is a new technology, which takes into account the characteristics of the palmprint line itself, such as its direction and width. Traditional methods mainly focus on local texture or edge features, and lack the ability to extract lines based on their inherent geometric characteristics. By doing so, WLE not only improves the clarity of palmprint line extraction, but also simplifies the data processing flow, making it a valuable contribution in the field of palmprint recognition.

(2) We propose a novel layered ViT (LViT) architecture. Unlike conventional models, LViT leverages the unique block structures of different palmprint images by fusing multiple block branches. This innovative approach not only boosts recognition accuracy but also cuts down training time, enabling it to achieve state-of-the-art performance in palmprint recognition.

(3) The proposed LViT architecture has good anti-noise generalization capability. We simulated some bad conditions in the actual situations (such as rain, storm and dust), and added salt-and-pepper noise, Gaussian noise and random occlusion to all the dataset. The overall recognition performance of our method clearly outperforms the performance of previous state-of-the-art methods.

(4) We provide a high-resolution palmprint database named XINHUA.

## RELATED WORK

Image line features retain important information about the shape of objects in the scene and play an important role in advanced tasks, including matching and recognition (*Davis, Rosenfeld & Agrawala, 1976*). Generally, palmprint recognition uses the palm of a person to identify or verify their identity (*Li et al., 2009*). Palmprint has many important features, such as principal lines, wrinkles, ridges, minutiae and textures. Among these features, the principal line, as one of the most obvious features of palmprints, has always been a focus research topic (*Wu, Wang & Zhang, 2002*). Usually, palmprints contain three principal lines: heart line, head line and life line (*Huang, Jia & Zhang, 2008*). If three smooth and noiseless principal lines can be extracted as feature images, it will lay a good foundation for fast matching and recognition for later. Therefore, extracting the principal lines of palmprint is crucial for palmprint recognition. Up to now, palmprint recognition has studied many algorithms for line detection and matching, including those based on palmprint principal lines and texture lines (some tiny short lines and

ridges), which are mainly divided into traditional methods and deep learning-based methods.

In traditional palmprint line extraction methods, detectors are generally designed manually to extract line features. Early studies mainly focused on basic feature detection and matching. For instance, *Wu, Wang & Zhang (2002)* designed a set of line detectors to assess the smoothness, connectivity, and width of lines, employing the Hausdorff distance for line matching. In the context of low-resolution palmprint recognition, *Wu et al. (2004)* developed a set of directional line detectors, along with a new automatic classification algorithm based on the width and thickness of palmprint principal lines.

As research progressed, some scholars began to focus on extracting principal lines from the characteristics of images. *Liu & Zhang (2005)* obtained the intensity relationship between palmprint lines by minimizing local image regions with similar brightness to each pixel, thus extracting the principal lines of palmprints. *Wu & Zhang (2006)* extracted line features in different directions and represented some fine lines using chain codes. During the matching stage, they performed matching based on the distances between points on the palmprint lines. *Liu, Zhang & You (2007)* adopted an isotropic nonlinear filter as a wide-line detector from the perspective of line width, achieving a relatively robust line extraction effect.

Meanwhile, methods based on transformation and retrieval gradually emerged. *Huang, Jia & Zhang (2008)* utilized the improved finite radon transform (MFRAT) to extract the principal line features of palmprints and employed a "point-to-region" approach for matching, achieving good results. *Jia et al. (2009)* proposed a fast palmprint retrieval scheme based on principal lines, leveraging the position and direction of key points on principal lines to retrieve palmprints, which achieved extremely fast retrieval results while ensuring good accuracy. *Li et al. (2009)* first used MFRAT to extract the preliminary principal lines of palmprints and then obtained the final refined principal lines through post-processing operations such as binarization and morphology.

Subsequent research further deepened the combination of preprocessing and feature optimization. *Li, Liu & Zhang (2010)* first preprocessed the images (such as median filtering, *etc.*), then refined the detected palmprint lines based on diversity and contrast, and used edge tracking to remove thin branches and short lines to obtain the principal lines of palmprints. *Yuan et al. (2011)* designed a dotted-link algorithm using the principal line tracking method, taking into account the specific direction of principal lines and the prior knowledge of valley edges. *Rotinwa-Akinbile, Aibinu & Salami (2011)* treated the principal lines of palmprints as edge detection, using Sobel operators in both horizontal and vertical directions to extract principal lines and applying discrete Fourier transform technology to calculate the distances from endpoint to endpoint for matching.

In addition, innovative methods based on morphology and edge detection continued to emerge. *Kalluri, Prasad & Agarwal (2012)* designed a set of wide principal line extractors, using morphological operators and grouping functions to eliminate noise. During the matching stage, they developed a matching algorithm based on pixel comparison to calculate the similarity between palmprints. *Di, Shi & Xu (2013)* first used Nibrack's

method to roughly segment the principal lines of palmprints, then applied the SUSAN operator to limit the range of principal lines as the localization result, and finally achieved accurate extraction of palmprint principal lines through the intersection of rough-segmentation and localization results. *Biradar (2013)* proposed using canny edge detection to extract principal line features, using Sobel masks to find the edge direction and gradient intensity of each pixel in the preprocessed image, and then tracking the edges. Finally, non-maximum edges were suppressed by identifying parallel edges and eliminating those with weaker gradient strength. *Bruno et al. (2014)* achieved a simple, efficient, and accurate method for extracting palmprint principal lines through image normalization, median filtering, average filtering, gray combined filtering, binarization, and post-processing.

In recent years, research directions have expanded towards multimodality and algorithm innovation. *Iula & Nardiello (2016)* proposed a biometric recognition method based on ultrasound. In the same year, *Ali, Yannawar & Gaikwad (2016)* used local entropy information and local variance for edge detection, exploring the potential of some classical edge operators (such as the Sobel operator, *etc.*) in extracting palmprint principal lines. *Sathish, Baskar & Kumar (2021)* used the Prewitt edge detector, Sobel operator, canny edge detector, Kirsch operator, and multi-scale edge detector to extract the features of palmprint lines.

In terms of algorithm optimization, recent studies have made remarkable progress. *Wang & Mariano (2024)* proposed a local ordinal code (LOC) using three common filters for multi-lines-directional filtering coding to overcome the high computational cost and low accuracy of traditional palmprint local recognition methods, introduced a dimension control factor for linear dimensionality reduction to enable large-scale retrieval, developed FFLOC for feature fusion. *Liao et al. (2024)* proposed boundary line calibration (BLC) and finger valley calibration (FVC) to tackle translational dislocations in the DLSP assisting graph for mobile palmprint recognition. By rotating samples, cropping specific regions, applying Gabor filters, and localizing key features, their method effectively improved recognition accuracy and user comfort. *Wang & Cao (2025)* proposed the bifurcation line direction coding (BLDC) method to overcome challenges in palmprint recognition like image variability and limitations of traditional single-line-feature-based methods. Using an improved Gabor filter for preprocessing and generating feature codes based on main direction subscripts.

Deep learning, as a powerful machine learning technique, is highly regarded in current scientific research and engineering applications. In recent years, with the continuous development and optimization of deep learning models, the applications of deep learning in various fields have become increasingly widespread and profound. Studies in COVID-19 early diagnosis (*Zivkovic et al., 2022*; *Jain et al., 2023*), geological disaster prediction (*Chen & Song, 2023*), financial market forecasting (*Mousapour Mamoudan et al., 2023*), and pedestrian detection (*Jain et al., 2023*) have all demonstrated the outstanding performance of combining deep learning with metaheuristic optimization algorithms in solving complex problems. These studies not only overcome the limitations of traditional algorithms but also provide new ideas and methods for future academic research and

practical applications. The popularity of deep learning is attributed to its enormous potential and continuous exploration in solving complex real-world problems. Some researchers have gradually turned to deep learning to perform palmprint line extraction. *Wang et al. (2013)* proposed a method to quickly extract palmprints using image data field and pulse-coupled neural network (PCNN). *Putra et al. (2021)* and other researchers added edge detection results to convolutional neural network data for training purposes. *Zhao et al. (2022)* proposed synthesizing training data by processing palmlines. These studies verify that deep learning can effectively perform palmprint line extraction. *Parulan, Borcelis & Linsangan (2024)* proposed an innovative dynamic image segmentation-based approach for palm line identification and analysis, using palm line images captured by a device as biometric data, implementing percentage error for statistical treatment. *Jia & Zhou (2024)* proposed UC-HRNet, a high-resolution network for palmprint principal line extraction, aiming to address the challenges in dense prediction tasks of semantic segmentation. Leveraging the parallel feature map preservation of HRNet and the effective feature fusion of UNet's U-shaped structure with skip connections, and using deep supervision for hierarchical feature representation. In recent years, cross-domain palmprint recognition is mostly used in transfer learning, aiming at bringing the knowledge learned in one domain to another domain to realize domain adaptation. *Shao, Zhong & Du (2019)* proposed PalmGAN to address cross-domain palmprint recognition by generating labeled fake images that reduce domain gaps while preserving identity information. *Ruan, Li & Qin (2024)* proposed LSFM, an efficient light style and feature matching method for cross-domain palmprint recognition, addressing the challenges of domain shifts and resource limitations. *Xin et al. (2024)* proposed a self-attention CycleGAN for cross-domain semi-supervised palmprint recognition, addressing challenges in contactless palmprint recognition with different devices and limited labeled data. In the future, with the emergence of more cross-domain methods, such as those based on generative adversarial networks (GANs) and self-attention mechanisms, cross-domain palmprint recognition is expected to significantly enhance accuracy and efficiency in real-world applications. Additionally, strategies that combine small amounts of labeled data with unsupervised learning will further drive the adoption of cross-domain palmprint recognition in practical settings, especially in areas like security authentication and smart access control.

## PALMPRINT DATASET

This study utilizes five palmprint datasets: PolyU II (*Bruno et al., 2014*), IIT Delhi (*Kumar & Shekhar, 2011*), XINHUA (*Wang & Mariano, 2024*), NTU-CP-V1 (*Matkowski, Chai & Kong, 2020*), and BJTU-V2 (*Chai, Prasad & Wang, 2019*). This section describes the characteristics, collection methods, and advantages of each dataset. Table 1 summarizes the key detail of the five palmprint datasets.

This work utilized PolyU II palmprint database sourced from http://www4.comp.polyu. edu.hk/~biometrics/, utilized IIT Delhi Touchless Palmprint Database sourced from https://www4.comp.polyu.edu.hk/~csajaykr/IITD/Database_Palm.htm,

**Table 1 Introduction to palmprint dataset.**

| Dataset | Attribute | | | |
| --- | --- | --- | --- | --- |
| | Is it a restricted environment | Number of collectors | Number of categories | Total number of images |
| PolyU II (2014) | Yes | 193 | 386 | 7,752 |
| IIT Delhi (2011) | Yes | 230 | 460 | 2,601 |
| XINHUA (2024) | Yes | 50 | 100 | 2,000 |
| NTU-CP-V1 (2020) | No | 328 | 655 | 2,478 |
| BJTU-V2 (2019) | No | 148 | 296 | 2,663 |

utilized XINHUA Palmprint Database sourced from https://github.com/HewelXX/Dataset/tree/main/XINHUA, utilized NTU-CP-v1 Palmprint Database sourced from https://github.com/matkowski-voy/Palmprint-Recognition-in-the-Wild, utilized BJTU-V2 Palmprint Database sourced from https://github.com/HewelXX/Dataset/tree/main/BJTU_V2.

## PolyU II

The PolyU II dataset is a widely used contact-based 2D palmprint database. The dataset comprises 386 palms from 193 participants, covering both left and right hands. The data collection was conducted in two phases with a two-months interval between, with approximately 10 palmprint images collected per phase in each, resulting in a total of 7,752 images. The PolyU II is renowned for its large sample size and distinct palmprint features, making it widely used in palmprint feature extraction and matching research. Its advantages lie in its abundant samples and well-standardized collection process, while its limitation stems from its sole focus on contact-based collection environments, which results in a lack of diversity in non-contact scenarios.

## IIT Delhi

The IIT Delhi dataset is a contact-based palmprint database comprising 2,601 images from 460 palms of 230 participants. For each palm, five to seven images were captured under different hand postures. The dataset includes both raw palmprint images and normalized, cropped images ($150 \times 150$ pixels), which facilitates research and comparison among feature extraction algorithms. The advantage of IIT Delhi dataset lies in its inclusion of diverse hand postures and normalized images, which makes it an ideal data source for comparative analysis of palmprint images under varying conditions.

## XINHUA

The XINHUA dataset is a high-resolution palmprint database developed specifically for this study, aiming to provide richer data support for palmprint recognition research. The dataset includes 2,000 palmprint images from 50 participants, comprising 41 males and nine females, all aged 20 to 30. The data collection was conducted in two phases, from January 2022 to April 2022, with each participant providing 10 images of their left hand and 10 images of their right hand during each phase.

The data was collected using an iPhone XR smartphone in an indoor setting, with fixed lighting conditions and a stable shooting distance of approximately 20 centimeters. During the collection process, participants were asked to spread their palms flat, avoid occlusion and excess shadows, and ensure a simple and clean background. The data are stored in high-resolution image format, facilitating the extraction of palmprint feature details. Using a smartphone as the collection device makes the dataset more relevant to real-world applications, improving its practicality and usability. A standardized collection protocol ensures consistency in data quality.

### NTU-CP-V1

The NTU-CP-v1 is a contactless palmprint database comprising 2,478 images from 655 palms of 328 participants. The participant pool is predominantly of Asian descent (including Chinese, Indian, and Malay), with a small number of Caucasians and Eurasians. The data collection was conducted in two phases, in everyday indoor environments in Singapore, with no strict posture requirements. The images were captured using Canon EOS 500D and Nikon D70s cameras, ensuring high image quality. The advantage of NTU-CP-v1 lies in its contactless collection, which more closely resembles real-world usage scenarios. Additionally, it covers a diverse range of ethnic backgrounds, providing significant support for cross-ethnic palmprint feature research.

### BJTU-V2

The BJTU-V2 dataset comprises 2,663 hand images from 148 volunteers (91 males and 57 females), with ages ranging from 8 to 73 years. The data collection was conducted in two phases, from November 2015 to December 2017, with each participant providing three to five images of their left hand and three to five images of their right hand during each phase. The images were captured using various smartphones (such as iPhone 6, Nexus 6P, Huawei Mate8, *etc.*) in both indoor and outdoor settings.

## DATA AUGMENTATION OPERATION

Data augmentation is a technology for artificially expanding the training dataset by generating more equivalent data from limited data. It is an effective means to overcome the shortage of training data, and it is widely used in various fields of deep learning. In this article, a WLE filter is proposed to perform preliminary line extraction on the original image, and the specific filter template is shown in Fig. 1. Due to the complexity of the legend, only one direction of the filter template is listed in Fig. 1A (template size is $13 \times 13$, in the actual extraction process, the template size is set to $35 \times 35$). The templates in Figs. 1B and 1C present simplified diagrams, in which Fig. 1B illustrates the left-hand template and Fig. 1C illustrates the right-hand template. It should be noted that the red curve in Fig. 1A has the same functional representation as the red curves in Figs. 1B and 1C, both of which denote the template direction.

The choice of left-hand and right-hand templates is based on both empirical observations and experimental validation. From an empirical perspective, the left-hand

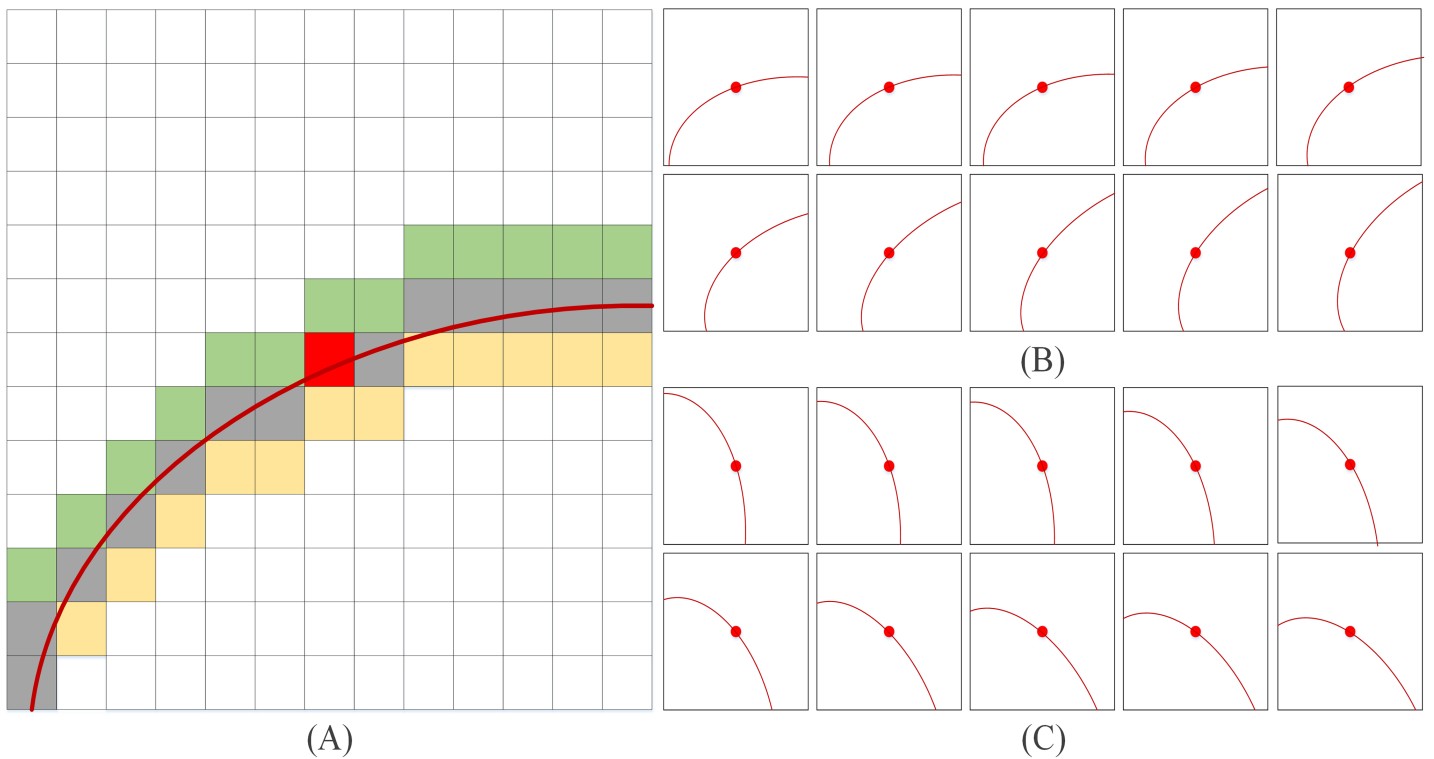

**Figure 1 WLE filter.** (A) Only one direction of the filter template (B) left-hand template (C) right-hand template.

and right-hand directions correspond to the natural orientations of most palmprint lines, ensuring that the templates align well with the dominant features. Experimentally, multiple template configurations were assessed he left-hand and right-hand templates exhibited superior performance in significantly improving line extraction accuracy and robustness. Furthermore, as illustrated in Fig. 1A, the width of template line direction is set to three pixels, as the principal palmprint lines typically have a width ranging from two to five pixels (*Jia, Huang & Zhang, 2008*).

When performing convolution filtering, WLE sums all pixel values within a region of three-pixel width. The specific equation is given by Eq. (1)

$$I_{WLE} = I * WLE \tag{1}$$

where WLE is defined as follows:

Given $Z_p = \{0, 1, \ldots, p-1\}$, where $p$ is a positive number representing the size of the grid, on a finite grid $Z^2_p$, the real-value function of WLE is given by Eq. (2), where k represents the width of the line, $L_k$ denotes the set of points that make up a line on the lattice $Z^2_p$, $f[x, y]$ is the gray value on the image $I$, $\rho$ is a limiting parameter to prevent the sum of gray values from exceeding 255, and $\rho$ ranges from 0 to 1. It is worth noting that before using WLE to perform convolution filtering, it is necessary to use mean reduction to preprocess the image, that is, $I = I' - mean(I')$.

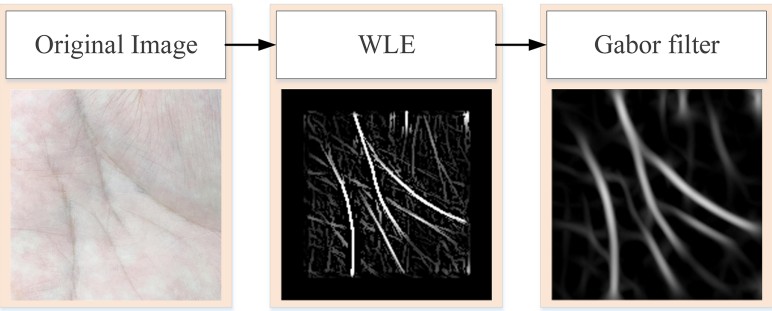

**Figure 2 Process of extracting pure palmprint principal line.**

$$WLE(k) = \rho \sum_{k=1}^{3} \sum_{(i,j) \in L_k} f[i,j]. \tag{2}$$

WLE only preliminarily screens the more obvious lines in the palm, such as some fine cross lines. To obtain the more obvious principal lines of the palmprint, further data preprocessing operations are needed. Figure 2 is the flowchart of obtaining the principal line of palmprint. First, the original palmprint image is processed by WLE filter, and then the filtered image is convolved by Gabor filter to remove any fine lines that remain after WLE processing.

## LAYERED VISUAL TRANSFORMER PARADIGM

Previously published material must be accompanied by written permission from the author and publisher. ViT's idea is to divide the image into blocks and send these blocks to the encoder, which inevitably leads to the question of whether all images are suitable for the same blocking strategy? The process of the model learning different things is both difficult and easy. Simple images are easy to identify and do not require dividing the image into multiple patches for training. However, dividing the image into multiple patches can lead to the shortcomings of long training time and insufficient precision. Therefore, this article introduces LViT, a Layered ViT model paradigm. The core idea is straightforward: the input image is divided into different numbers of patches for separate training, and the linear results of each patch number are summed and fused to output. Figure 3 illustrates the architecture of the LViT paradigm (here using only two blocking methods as examples). LViT extends the transformer architecture to enable multiple classification token head outputs. To distinguish the number of blocks between multiple modalities, we expanded the original Transformer architecture to include one-dimensional multi-patch embedding, which marks each blocking method and enhances network's generalization ability.

Here, ViT (*Dosovitskiy et al., 2021*) is taken as an example and its formula is followed, which is supplemented and explained in detail. In Eq. (3), $Z$ represents the output set of all different blocks, where $z_i$ represents the output of each block scheme (Eq. (4)). Here, $M$
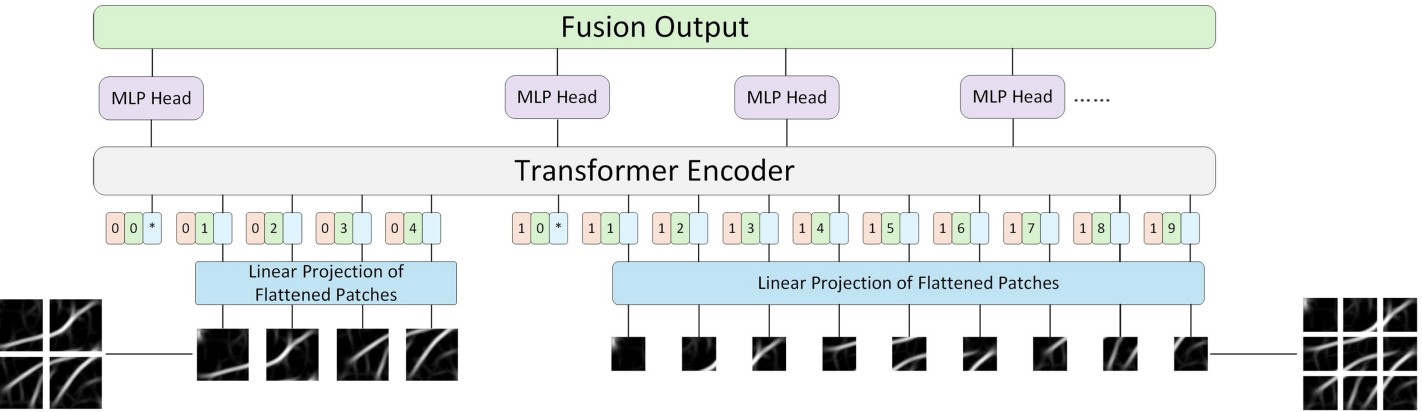

**Figure 3 LViT papadigm.**

represents the number of types of block schemes, $N$ represents the number of blocks in each block scheme, $x_{multi-patch}$ and $x_{class}$ are two learnable embeddings, representing the block-level and category-level embeddings, respectively, and $x_p$ represents the sequence of flattened 2D patches, $\mathbb{R}$ represents real numbers, $P^2$ represents the resolution of each image patch, $C$ represents the number of image channels, and $D$ represents the dimension of Transformer mapping layer.

$$Z = [z_0;\ z_1;\ z_2;\ \ldots;\ z_M]. \tag{3}$$

$$z_i = \left[ x_{multi-patch};\ x_{class};\ x_p^1 E;\ x_p^2 E; \ldots;\ x_p^N E; \right] + E_{pos},$$

$$E \in \mathbb{R}^{(P^2 \cdot C) \times D},\ E_{pos} \in \mathbb{R}^{(N+1) \times D},\ i = 1, \ldots, M. \tag{4}$$

Next, in order to obtain the final classification result of each block scheme, this article did not directly use the concatenated result in Eq. (3) as the input in the Encoder part, but sent each block scheme into the Encoder as a different input for separate training. This approach aligns with the original encoder parameter settings of ViT, thereby enhancing training efficiency for future transfer learning applications. Equations (5), (6) and (7) are the multi-headed self-attention (MSA), multilayer perceptron (MLP) and LayerNorm (LN) components of the Encoder, respectively. The superscript 1 in $z_L^1$ in Eq. (7) denotes the dimension index of the class token. In this algorithm, the class token is placed in the first dimension, while multi-patch embedding is in the 0th dimension, which is different from ViT where the class token is in the 0th dimension. $L$ denotes the maximum number of layers in the Encoder.

$$z_l' = MSA(LN(z_{l-1})) + z_{l-1}, \quad l = 1, \cdots, L. \tag{5}$$

$$z_l = MLP(LN(z_l')) + z_l' \quad l = 1, \cdots, L. \tag{6}$$

$$y = LN(z_L^1). \tag{7}$$

After obtaining each block scheme, this article fuses the results of each scheme to make the final prediction. This approach is referred to as LViT. Specifically, in the corresponding fusion strategy, two fusion methods, namely "sum fusion" and "maximum fusion", are

adopted to fuse the matching value layer. Equation (8) represents the "summation fusion" method, where N is the number of block types, and *S* denotes the output score of each block. Equation (9) represents the for "maximum fusion" method.

$$LViT_{Sum} = Sum(S_{block1}, S_{block2}, \ldots, S_{blockN}) = \frac{1}{N} * (S_{block1} + S_{block2} + \ldots + S_{blockN}). \qquad (8)$$

$$LViT_{Max} = Max(S_{block1}, S_{block2}, \ldots, S_{blockN}). \qquad (9)$$

## DISCUSSION AND ANALYSIS OF EXPERIMENTAL RESULTS

In this section, the Transformer architectures used are the existing backbone networks, namely ViT, Conformer (*Peng et al., 2021*), PVT-V2 (*Wang et al., 2022*) and ConvMixer (*Trockman & Kolter, 2022*). ViT is the first successful attempt to introduce the Transformer into the vision field, setting a precedent for vision Transformers. ViT converts image data into sequence data and feeds it into the standard Transformer encoder to achieve higher recognition accuracy. Conformer is based on the feature coupling unit (FCU), which fuses local features and global representations under different resolutions in an interactive fashion. Conformer adopts a concurrent structure to maximize the retention of local features and global representations. PVT-V1 (*Wang et al., 2021*) is the first pyramid-structured Transformer model, which proposes a hierarchical Transformer with four stages, demonstrating that a pure Transformer backbone can be as universal as the CNN backbone. PVT-V2, on the other hand, introduces overlapping patch embedding, convolutional feed-forward, and linear spatial reduction attention to improve the recognition accuracy and reduce the computational complexity.

In verify the effectiveness of the proposed method this chapter, this section conducts experimental tests from the following three aspects. (a) Baseline experiments, referring to the training results of the original dataset using various backbone networks; (b) Comparative experiments using WLE data augmentation; (c) Experimental results of LViT paradigm.

In this study, the training strategies and parameter settings for deep models were based on the approach described by (*Zivkovic et al., 2022*), whereas the experimental settings for traditional methods followed those of (*Dosovitskiy et al., 2021*). For deep model training, the training and testing sets were strictly divided according to the collection phases, using data from the first phase as the training set and data from the second phase as the testing set. Since the IIT Delhi dataset has only one collection phase, the first collected sample from each class was used as the training set, and the remaining samples were used as the testing set. Regarding the image size, images from different databases varied significantly. The original palmprint image size ranges from $128 \times 128$ pixels to $2,000 \times 2,000$ pixels. To ensure compatibility with our models and maintain computational efficiency, we resized all images in this dataset to a fixed size of $224 \times 224$ pixels using the bilinear interpolation method. A detailed summary of these division strategies for all databases is presented in Table 2 for easy reference. For deep learning methods, the batch size for training was set to 4, the learning rate to $5 \times 10^{-5}$, the optimizer to AdamW

**Table 2 The division strategies for all databases.**

| Dataset | Train number of training sets | Test number of training sets | Original palmprint image size | Input the palmprint image size of the model |
|---|---|---|---|---|
| PolyU II | 3,889 | 3,863 | 128 × 128 | 224 × 224 |
| IIT Delhi | 459 | 2,237 | 150 × 150 | 224 × 224 |
| XINHUA | 1,000 | 1,000 | 2,000 × 2,000 | 224 × 224 |
| NTU-CP-V1 | 1,304 | 1,086 | About 500 × 500 resolution | 224 × 224 |
| BJTU-V2 | 1,341 | 1,322 | About 1,000 × 1,000 resolution | 224 × 224 |

**Table 3 The main evaluation index of palmprint recognition system.**

| Evaluation index | Abbreviation | Description |
|---|---|---|
| False rejection rate | FRR | The proportion of genuine being incorrectly rejected by the classifier. |
| False acceptance rate | FAR | The proportion of impostors mistakenly judged as accepted by the classifier. |
| Genuine acceptance rate | GAR | The concept of GAR is opposite to FRR, with a value of 1-FRR. |
| Receiver operating characteristic | ROC | The receiver operating characteristic intuitively reflects the balance relationship between GAR and FAR at different thresholds of the recognition algorithm, where the horizontal axis is FAR and the vertical axis is GAR. |
| Equal error rate | EER | The value of the ROC curve when FRR and FAR are equal. |
| Accuracy recognition rate | ARR | The ratio of correctly classified samples to total samples |

**Table 4 Baseline experimental results of each dataset in the restricted environment.**

| Methods | Patch size | PolyU II | | IIT Delhi | | XINHUA | |
|---|---|---|---|---|---|---|---|
| | | ARR | EER | ARR | EER | ARR | EER |
| ViT-B | 16 | 87.78 | 7.1427 | 76.64 | 14.3725 | 38.90 | 45.3584 |
| | 32 | 80.09 | 10.2715 | 70.15 | 20.4975 | 35.80 | 49.2105 |
| Conformer-B | 16 | 91.98 | 6.1325 | 81.19 | 9.3667 | 46.20 | 37.3344 |
| | 32 | 86.35 | 7.4487 | 74.22 | 16.3562 | 38.60 | 46.7724 |
| PVT-V2-B5 | 4 | 95.91 | 3.7245 | 91.78 | 5.9733 | 71.50 | 19.2238 |
| | 8 | 95.86 | 3.7326 | 91.36 | 6.0445 | 73.00 | 18.1582 |
| | 16 | 95.81 | 3.8046 | 89.96 | 6.8328 | 71.50 | 19.2674 |
| | 32 | 96.33 | 3.3026 | 90.33 | 6.7544 | 77.30 | 13.9044 |
| ConvMixer | 7 | 99.82 | 0.7526 | 92.89 | 5.7742 | 69.60 | 20.8743 |
| | 14 | 99.85 | 0.6344 | 89.47 | 6.9804 | 63.80 | 23.5576 |

(*Loshchilov & Hutter, 2018*), and data augmentation was implemented using RandAugment (*Cubuk et al., 2020*).

Generally speaking, the system performance evaluation criteria of palmprint recognition algorithm are shown in Table 3. Accuracy recognition rate (ARR), equal error rate (EER), genuine acceptance rate (GAR), false acceptance rate (FAR), receiver operating

**Table 5 Baseline experimental results of each dataset in the unrestricted environment.**

| Methods | Patch size | NTU-CP-V1 | | BJTU-V2 | |
|---|---|---|---|---|---|
| | | ARR | EER | ARR | EER |
| ViT-B | 16 | 72.99 | 18.2326 | 63.90 | 23.4826 |
| | 32 | 70.70 | 20.3879 | 60.05 | 24.5527 |
| Conformer-B | 16 | 78.39 | 12.0726 | 75.13 | 15.5475 |
| | 32 | 72.44 | 18.4275 | 70.25 | 20.4431 |
| PVT-V2-B5 | 4 | 83.24 | 9.3327 | 86.18 | 8.1022 |
| | 8 | 83.79 | 9.1872 | 84.34 | 9.1786 |
| | 16 | 85.44 | 8.3976 | 85.51 | 8.4725 |
| | 32 | 84.71 | 9.0547 | 85.76 | 8.3720 |
| ConvMixer | 7 | 84.89 | 9.0128 | 89.03 | 6.6547 |
| | 14 | 84.98 | 8.9744 | 87.35 | 7.1438 |

**Table 6 Data augmentation experiment results of each dataset in the restricted environment.**

| Methods | Patch size | PolyU II | | IIT Delhi | | XINHUA | |
|---|---|---|---|---|---|---|---|
| | | ARR | EER | ARR | EER | ARR | EER |
| ViT-B | 16 | 96.32 | 3.3524 | 84.74 | 9.0326 | 69.00 | 21.0745 |
| | 32 | 92.51 | 5.8236 | 76.84 | 14.3375 | 68.70 | 21.2427 |
| Conformer-B | 16 | 92.07 | 5.9045 | 85.07 | 8.5022 | 71.20 | 19.5585 |
| | 32 | 87.23 | 7.2032 | 78.57 | 12.0163 | 63.80 | 23.6042 |
| PVT-V2-B5 | 4 | 96.01 | 3.6355 | 92.33 | 5.7459 | 78.10 | 12.8741 |
| | 8 | 96.06 | 3.6218 | 94.20 | 4.5722 | 76.90 | 15.3670 |
| | 16 | 96.37 | 3.2883 | 92.51 | 5.7120 | 79.40 | 11.4032 |
| | 32 | 96.96 | 3.2047 | 93.64 | 5.0773 | 76.90 | 15.3528 |
| ConvMixer | 7 | 99.87 | 0.5833 | 93.42 | 5.1844 | 79.60 | 11.4427 |
| | 14 | 99.87 | 0.6028 | 89.87 | 6.5035 | 77.30 | 13.8725 |

characteristic (ROC) will be used as experimental evaluation indexes in the following experiments.

## WLE data augmentation experiment

This section evaluates the experiments with and without WLE data augmentation on five datasets, including three palmprint datasets in restricted environment and two datasets in unrestricted environment respectively. The selected evaluation models included ViT, Conformer, PVT-V2 and ConvMixer. But not limited to these evaluation models, any ViTs model can implement the algorithmic paradigms in this chapter. It should be added that in the current well-established deep learning methods, employing basic data augmentation techniques, such as rotation and cropping, has become a necessary operation. This subsection aims exclusively to demonstrate the effectiveness of the WLE data augmentation method and does not involve comparisons with other non-deep learning augmentation methods (*Cubuk et al., 2020*). Table 4 presents the baseline experiments of

**Table 7 Data augmentation experiment results of each dataset in the unrestricted environment.**

| Methods | Patch size | NTU-CP-V1 | | BJTU-V2 | |
|---|---|---|---|---|---|
| | | ARR | EER | ARR | EER |
| ViT-B | 16 | 75.02 | 16.9456 | 80.90 | 10.0302 |
| | 32 | 74.90 | 17.0188 | 76.48 | 15.6884 |
| Conformer-B | 16 | 79.44 | 11.3356 | 79.73 | 11.2562 |
| | 32 | 74.26 | 17.2163 | 72.18 | 18.8629 |
| PVT-V2-B5 | 4 | 85.99 | 8.1320 | 86.52 | 8.0421 |
| | 8 | 84.61 | 9.1128 | 85.76 | 8.3546 |
| | 16 | 85.53 | 8.3853 | 86.60 | 8.0225 |
| | 32 | 85.43 | 8.4015 | 86.01 | 8.1125 |
| ConvMixer | 7 | 85.35 | 8.4726 | 89.11 | 6.6325 |
| | 14 | 85.26 | 8.5877 | 87.69 | 7.0844 |

**Table 8 LViT experiment results without WLE data augmentation.**

| Methods | mode | PolyU II | | IIT Delhi | | XINHUA | | NTU-CP-V1 | | BJTU-V2 | |
|---|---|---|---|---|---|---|---|---|---|---|---|
| | | ARR | EER | ARR | EER | ARR | EER | ARR | EER | ARR | EER |
| L-ViT | S | 90.44 | 5.9034 | 80.35 | 11.4452 | 53.90 | 33.1982 | 74.06 | 17.8621 | 72.30 | 16.8567 |
| | M | 91.02 | 5.7214 | 80.88 | 11.0321 | 56.20 | 31.2283 | 74.22 | 17.5523 | 73.20 | 15.2083 |
| L-Conformer | S | 92.00 | 6.0912 | 83.45 | 7.9034 | 58.90 | 28.3376 | 78.93 | 11.8214 | 77.29 | 13.2904 |
| | M | 92.02 | 6.0004 | 83.92 | 7.1238 | 59.40 | 27.3592 | 79.11 | 11.6218 | 78.44 | 12.8703 |
| L-PVT-V2 | S | 96.48 | 3.2896 | 92.34 | 5.7213 | 78.34 | 12.9886 | 85.51 | 8.3614 | 86.22 | 8.0823 |
| | M | 96.56 | 3.2455 | 93.06 | 5.0662 | 79.02 | 11.9232 | 85.78 | 8.2514 | 86.28 | 8.0546 |
| L-ConvMixer | S | 99.85 | 0.6134 | 92.95 | 5.6713 | 74.50 | 15.3893 | 85.12 | 8.5893 | 89.08 | 6.6507 |
| | M | 99.87 | 0.6016 | 93.11 | 5.5523 | 76.20 | 14.3328 | 85.33 | 8.5012 | 89.11 | 6.6425 |

each dataset (PolyU II, IIT Delhi and XINHUA) in the restricted environment, while Table 5 presents the baseline experiments of each dataset (NTU-CP-V1 and BJTU-V2) in the unrestricted environment.

Next, the palmprint images after WLE data augmentation were evaluated to verify the effectiveness of WLE data augmentation. Table 6 presents the experimental results after data augmentation for the datasets PolyU II, IIT Delhi and XINHUA in the restricted environment, while Table 7 presents the experimental results after data Augmentation for the datasets NTU-CP-V1 and BJTU-V2 in the unrestricted environment. The experimental results show that the ViT model with WLE data augmentation exhibits stronger generalization ability. The recognition rate and EER have been further improved.

## LViT experiment

This section evaluates the LViT paradigm. In this article, the backbone network name is renamed by adding a prefix identifier. In the experimental results, S denotes "summation fusion" and M denotes "maximum fusion". In this section, two types of LViT ablation

**Table 9 LViT experiment results with WLE data augmentation.**

| Methods | mode | PolyU II | | IIT Delhi | | XINHUA | | NTU-CP-V1 | | BJTU-V2 | |
|---|---|---|---|---|---|---|---|---|---|---|---|
| | | ARR | EER | ARR | EER | ARR | EER | ARR | EER | ARR | EER |
| L-ViT | S | 96.68 | 3.2714 | 85.03 | 9.0211 | 71.20 | 19.5445 | 78.45 | 12.6875 | 81.30 | 9.9468 |
| | M | 96.96 | 3.1825 | 86.14 | 8.2875 | 71.60 | 19.1528 | 78.76 | 12.5721 | 81.45 | 9.8723 |
| L-Conformer | S | 92.44 | 5.7546 | 86.54 | 8.1323 | 73.30 | 18.5465 | 79.88 | 11.0238 | 80.65 | 10.1872 |
| | M | 93.17 | 5.4635 | 86.93 | 8.0144 | 73.30 | 18.4432 | 80.15 | 10.8652 | 81.20 | 9.9833 |
| L-PVT-V2 | S | 97.15 | 3.0421 | 94.47 | 4.1732 | 78.60 | 12.2874 | 86.98 | 7.8833 | 87.38 | 7.3625 |
| | M | 97.44 | 2.9328 | 94.98 | 3.9828 | 79.60 | 11.3678 | 87.44 | 7.4902 | 87.60 | 7.1256 |
| L-ConvMixer | S | 99.93 | 0.5624 | 93.47 | 5.1221 | 79.80 | 11.4279 | 89.35 | 6.4805 | 89.23 | 6.5546 |
| | M | 99.93 | 0.5525 | 93.65 | 5.0136 | 80.10 | 10.7454 | 89.78 | 6.1736 | 89.76 | 6.2045 |

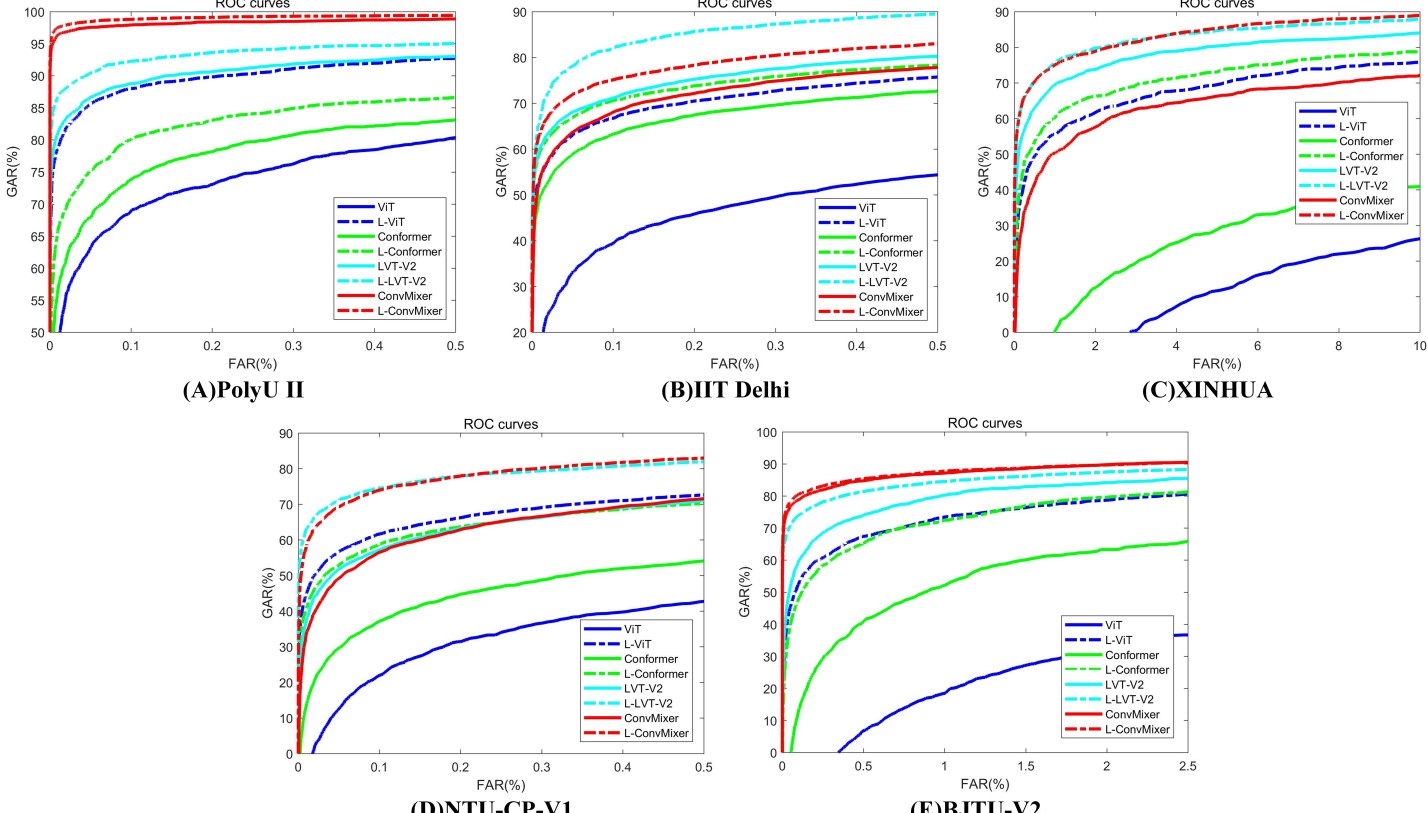

**Figure 4 ROC characteristic curve after WLE augmentation operation.** (A) PolyU II (B) IIT Delhi (C) XINHUA (D) NTU-CP-V1 and (E) BJTU-V2.

experiments were conducted, one without WLE data augmentation and the other with WLE data augmentation.

Table 8 presents the experimental results for LViT without WLE data augmentation. It can be seen from Tables 7 and 8 that the augmentation amplitude of WLE is better than that of LViT, that is, the influence of WLE on the model is better than the definition of the

**Table 10 Comparative experimental results.**

| Methods | PolyU II | | IIT Delhi | | XINHUA | | NTU-CP-V1 | | BJTU-V2 | |
|---------|------|------|------|------|------|------|------|------|------|------|
| | ARR | EER | ARR | EER | ARR | EER | ARR | EER | ARR | EER |
| CompC (*Zhang et al., 2003*) | 100 | 0.0513 | 89.78 | 5.4762 | 79.20 | 10.6030 | 89.47 | 6.7741 | 87.77 | 6.6823 |
| OrdinalC (*Sun et al., 2005*) | 100 | 0.0497 | 88.42 | 6.2285 | 73.10 | 12.0343 | 87.55 | 7.5996 | 88.61 | 5.4721 |
| RLOC (*Jia, Huang & Zhang, 2008*) | 100 | 0.0521 | 87.69 | 6.3400 | 67.50 | 15.1470 | 86.81 | 7.7674 | 87.52 | 6.0517 |
| LLDP (*Luo et al., 2016*) | 100 | 0.0517 | 91.67 | 4.3437 | 76.70 | 11.2780 | 88.74 | 7.7645 | 91.88 | 5.3637 |
| EEPNet (*Jia et al., 2022*) | 99.67 | 0.5907 | 87.65 | 7.9846 | 68.40 | 16.1294 | 85.07 | 7.9846 | 82.75 | 9.7481 |
| CCNet (*Yang et al., 2023a*) | 99.97 | 0.1554 | 31.40 | 40.7147 | 78.30 | 14.7012 | N/A | N/A | 91.04 | 5.9900 |
| CO3Net (*Yang et al., 2023b*) | 99.74 | 0.2814 | 64.72 | 24.3310 | 72.80 | 15.1947 | 70.97 | 15.5140 | 87.52 | 7.7089 |
| L-ViT (Proposed) | 96.96 | 3.1825 | 86.14 | 8.2875 | 71.60 | 19.1528 | 78.76 | 12.5721 | 81.45 | 9.8723 |
| L-Conformer (Proposed) | 93.17 | 5.4635 | 86.93 | 8.0144 | 73.30 | 18.4432 | 80.15 | 10.8652 | 81.20 | 9.9833 |
| L-PVT-V2 (Proposed) | 97.44 | 2.9328 | 94.98 | 3.9828 | 79.60 | 11.3678 | 87.44 | 7.4902 | 87.60 | 7.1256 |
| L-ConvMixer (Proposed) | 99.93 | 0.5525 | 93.65 | 5.0136 | 80.10 | 10.7454 | 89.78 | 6.1736 | 89.76 | 6.2045 |

framework. Table 9 presents the experimental results of LViT with WLE data augmentation. The effectiveness of WLE is further demonstrated in Tables 8 and 9, where the accuracy of ViT and Conformer shows significant improvement. Among them, the M fusion method achieves higher accurate than the S fusion method, because the fusion score takes the maximum score and often can obtain the best selection parameters. The ROC characteristic curve after WLE augmentation operation is illustrated in Fig. 4.

## Contrast experiment

In this section, we compare four traditional methods including CompC (*Zhang et al., 2003*), OrdinalC (*Sun et al., 2005*), RLOC (*Jia, Huang & Zhang, 2008*) and LLDP (*Luo et al., 2016*) and three methods based on deep learning including EEPNet (*Jia et al., 2022*), CCNet (*Yang et al., 2023a*) and CO3Net (*Yang et al., 2023b*) to demonstrate the feasibility and effectiveness of LViT. For the sake of fairness, WLE data augmentation operation is used in the comparative experiments of the three deep models, and no other gain operation is performed on the data and models.

Table 10 presents the results of comparative experiments. Due to the manual design of feature extractor, traditional manual methods can often achieve good results in some specific situations, such as PolyU II and NTU-CP-V1. Specifically, LLDP achieved the best performance on BJTU-V2 dataset. However, this does not mean that models based on the LViT paradigm are slightly worse on some specific datasets. This article aims to explain the beneficial effect of the LViT paradigm, but does not dig deep into the benefits brought by model architecture. In addition, it can be seen that the three palmprint recognition methods based on deep learning usually have poor performance. This is because the palmprint dataset is a small sample data, and it is often difficult to identify key features for the model trained from scratch. It is worth noting that CCNet achieves good recognition performance on PolyU II and BJTU-V2 datasets. The images in the NTU-CP-V1 dataset have relatively low resolution, and the sample size for each class is limited. These factors

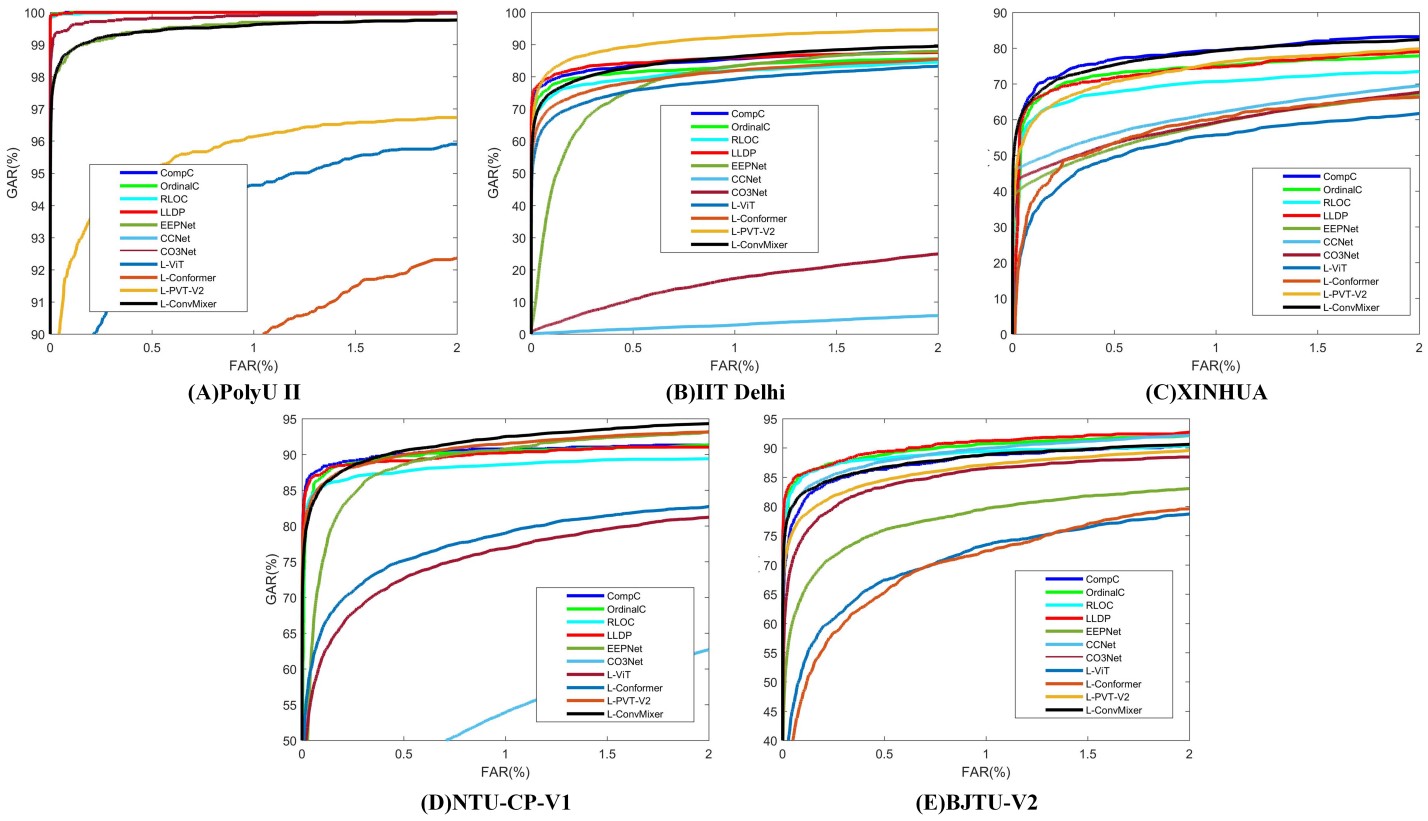

**Figure 5** The corresponding ROC characteristic curve of a comparative experiment. (A) PolyU II (B) IIT Delhi (C) XINHUA (D) NTU-CP-V1 and (E) BJTU-V2.

cause CCNet's receptive field to fail to effectively capture key features, which in turn affects its training convergence. A ROC characteristic curve of a comparative experiment is illustrated in Fig. 5.

## Anti-noise experiment

In real life, there are often worse situations, such as rain, storm, dust, and so on. In these extreme cases, deep learning methods can often exhibit a certain anti-noise capability. However, traditional manual palmprint recognition methods are affected by environmental and terrain factors, leading to reduced recognition. This section simulates several special scenarios and adds noise to palmprint data to verify the effectiveness and generalization ability of the LViT model.

There are three types of preprocessing on palmprint data: salt-and-pepper noise, Gaussian noise and random occlusion. Figure 6 illustrates the example after adding noise, where Fig. 6A illustrates the original image, and Figs. 6B, 6C and 6D illustrate the processing results of salt-and-pepper noise, Gaussian noise, and random occlusion. Table 11 presents the experimental results after noise processing, respectively. The corresponding ROC curve of the anti-noise datasets is illustrated in Fig. 7.

From the above experimental results, we can draw the following conclusions.

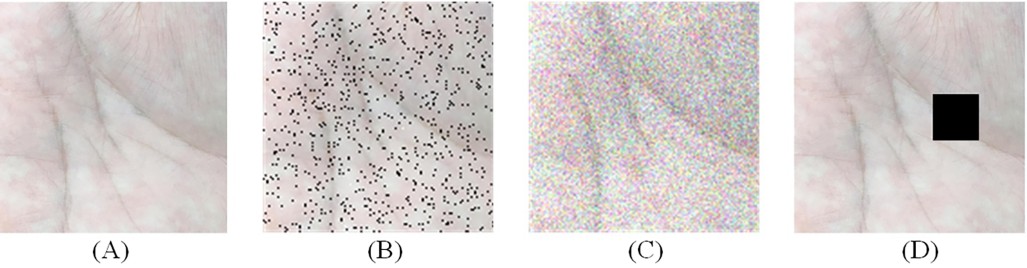

**Figure 6 Add noise processing.** (A) Original image (B) salt-and-pepper noise (C) Gaussian noise and (D) random occlusion.

**Table 11 Experimental results of noise.**

| Methods | PolyU II | | IIT Delhi | | XINHUA | | NTU-CP-V1 | | BJTU-V2 | |
|---|---|---|---|---|---|---|---|---|---|---|
| | ARR | EER | ARR | EER | ARR | EER | ARR | EER | ARR | EER |
| CompC (*Zhang et al., 2003*) | 91.66 | 2.5626 | 84.14 | 6.7584 | 75.80 | 12.6367 | 85.81 | 7.6930 | 56.86 | 28.2909 |
| OrdinalC (*Sun et al., 2005*) | 91.34 | 2.1021 | 82.53 | 8.0058 | 72.13 | 12.6926 | 84.91 | 8.2385 | 56.49 | 19.2576 |
| RLOC (*Jia, Huang & Zhang, 2008*) | 93.70 | 2.1130 | 78.91 | 9.4112 | 69.13 | 15.0854 | 83.60 | 9.3808 | 51.85 | 32.6343 |
| LLDP (*Luo et al., 2016*) | 84.59 | 7.2858 | 81.52 | 11.3362 | 71.40 | 12.4132 | 80.44 | 10.1568 | 58.22 | 26.5823 |
| EEPNet (*Jia et al., 2022*) | 97.44 | 1.4782 | 81.47 | 8.8326 | 60.47 | 17.3678 | 82.33 | 9.5523 | 66.92 | 15.5428 |
| CCNet (*Yang et al., 2023a*) | 99.44 | 0.9725 | N/A | N/A | 75.97 | 12.1136 | 72.80 | 11.4782 | N/A | N/A |
| CO3Net (*Yang et al., 2023b*) | 97.87 | 1.4033 | 58.37 | 18.3925 | 74.87 | 12.3217 | 62.00 | 16.9084 | 50.71 | 33.2165 |
| L-ViT (Proposed) | 93.27 | 2.1896 | 76.57 | 12.4426 | 68.87 | 15.9838 | 78.24 | 11.0236 | 57.25 | 25.3416 |
| L-Conformer (Proposed) | 91.31 | 2.6325 | 80.65 | 10.3527 | 73.36 | 12.6527 | 85.91 | 7.5893 | 46.67 | 35.7724 |
| L-PVT-V2 (Proposed) | 95.99 | 1.8274 | 86.15 | 6.2902 | 75.80 | 12.0445 | 87.53 | 6.7238 | 65.83 | 19.2863 |
| L-ConvMixer (Proposed) | 99.53 | 0.8722 | 90.58 | 5.2568 | 77.40 | 11.3823 | 86.00 | 7.5560 | 67.39 | 17.3348 |

(1) The results of L-ViT are outperform than traditional manual methods in most cases, because ViT benefits from data augmentation, which increases the anti-noise and generalization ability. However, traditional methods are greatly influenced by the environment and are usually suitable for discrimination in a standardized indoor environment;

(2) No traditional method can achieve the best results. Similarly, no L-ViT result is the best in all datasets. This also reflects that the influence of data on the model is uncontrollable. However L-ViT can still mitigate this influence;

(3) The fact that CCNet can be retrained on the NTU-CP-V1 dataset indicates that the data volume determines the model's generalization ability, which aligns with the results in Table 10. Additionally, the IIT Delhi dataset becomes untrainable on the CCNet model when the data volume is increased, possibly due to the extremely limited number of training samples (only one sample);

(4) Traditional manual methods often demonstrate inferior anti-noise performance compared to deep learning-based approaches, because traditional methods rely on

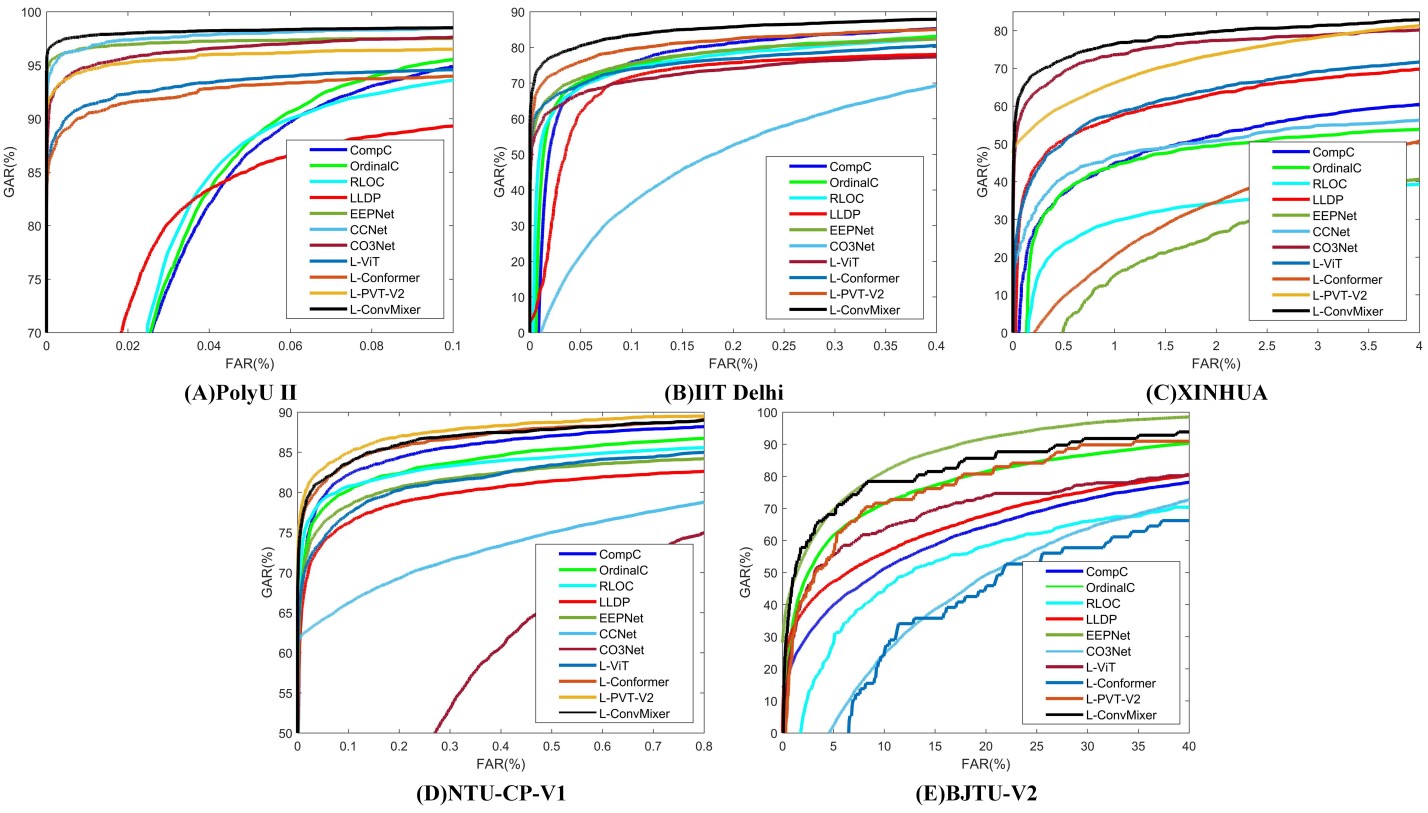

**Figure 7** **The corresponding ROC curve of the anti-noise datasets.** (A) PolyU II (B) IIT Delhi (C) XINHUA (D) NTU-CP-V1 and (E) BJTU-V2.

manually designed descriptors, and the feature distribution of the original image changes in an uncertain noise environment;

(5) The experimental results of L-ConvMixer are generally superior compared to other methods, particularly on PolyU II and IIT Delhi datasets.

## Discussion and analysis

According to the previous experimental results, the feasibility of the LViT paradigm has been verified, and WLE data augmentation has played a significant role as well. This section focuses on the training time. Since LViT inputs many different patches into the model, it inevitably increases the overall Flops and MAdd of the model. But at the same time, it brings a shorter training time, that is, the number of training iterations is significantly reduced, and the convergence speed of the model is further improved. Figure 8 illustrates the loss function trajectory (with WLE) of each LViT method on the PolyU II dataset. It can be seen that LViT achieves faster convergence with fewer iterations, and its loss function is relatively smooth.

For real-time applications, computational complexity is an important factor to determine the processing speed. In the process of large-scale manual authentication, traditional methods often need to extract features online and match them after processing. Because LViT is based on a deep learning model, real-time matching can be achieved as

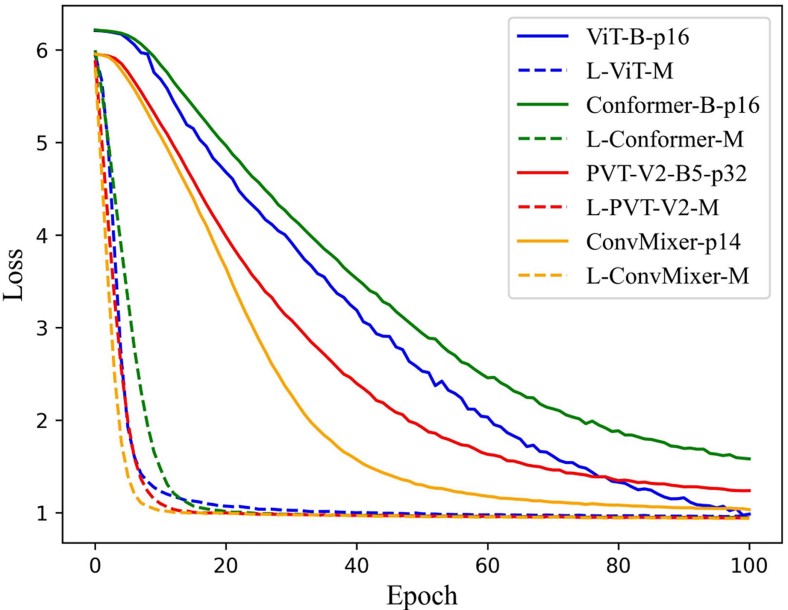

**Figure 8** **Loss function on PolyU II dataset.**

**Table 12** **Performance comparison of deep model.**

| CompC | Flops (G) | MAdd (M) | Memory (M) |
|---|---|---|---|
| CCNet (*Yang et al., 2023a*) | 0.257 | 0.514 | 5.89 |
| CO3Net (*Yang et al., 2023b*) | 0.11 | 0.22 | 3.07 |
| EEPNet (*Jia et al., 2022*) | 0.391 | 0.775 | 49.66 |
| L-ViT (Proposed) | 0.4 | 0.8 | 293.12 |
| L-Conformer (Proposed) | 26.73 | 53.31 | 818.68 |
| L-PVT-V2 (Proposed) | 1.44 | 2.85 | 315.40 |
| L-ConvMixer (Proposed) | 51.29 | 102.45 | 738.01 |

long as the model training is completed. In addition, PolyU II data set is a controlled environment database, so traditional methods usually have better results, while IIT Delhi, XINHUA, NTU-CP-V1 and BJTU are more complex databases, so deep learning methods typically achieve better performance. Compared with other depth palmprint methods, the determining conditions of computational complexity usually include Flops, MAdd and Memory. In order to be fair, the input of all depth models is scaled to 224 * 224. Table 12 presents the performance comparison of deep models.

As shown in Table 12, LViT has higher Flops, MAdd and memory size than CCNet, CO3Net and EEPNet, which is consistent with expectations. LViT divides the image into multiple patches for processing, which results in higher computational complexity. The self-attention mechanism needs to calculate the attention scores between all patches, and the embedding of each patch requires matrix multiplication and nonlinear transformations, which increases the computational burden. LViT contains a large number

of layers, which are used to capture complex image features, resulting in increased computational and memory consumption.

The complexity of LViT mainly depends on two aspects: the resolution of the input image and the sequence length of the image (the number of patches). Generally speaking, assuming that the resolution of the input image is $H \times W$, and the size of each patch is $P \times P$, then the image is divided into $(H \times W)/P^2$ patches. Assuming that the sequence length is N, LViT uses a multi-layer self-attention mechanism (self-attention), and the complexity is roughly $O(N^2 \times D)$, where D is the dimension of the vector representation of each patch. In each Transformer block, in addition to the self-attention, it also includes a fully connected multi-layer perceptron (MLP) layer. The complexity of the MLP layer is $O(N \times M^2)$, where M is the dimension of the hidden layer in the MLP layer. Therefore, the overall complexity of LViT can usually be expressed as $O((H \times W)/P^2 \times N^2 \times D + (H \times W)/P^2 \times N \times M^2)$.

Despite these higher computational costs, LViT has demonstrated superior performance in classification tasks, indicating that the benefits of its architectural design outweigh the associated computational overhead. Efforts to optimize LViTs' efficiency through techniques such as efficient attention mechanisms, model distillation, and pruning are ongoing research areas aimed at reducing their computational requirements while maintaining high performance. Recently, some lightweight ViTs (*Anasosalu Vasu et al., 2023*; *Mehta & Rastegari, 2022*; *Wang et al., 2024*) also provide the possibility of LViT extension, and the ViT-based methods will gradually improve performance to adapt to various devices. In addition, binary network and hash retry technology are new directions of consideration, which can map high-dimensional image data into low-dimensional binary coding space, reduce data storage and computational costs, and improve retrieval efficiency.

LViT finds the optimal model through various blocking strategies, which also have guiding role for non-Transformer-based neural networks. For example, it can use multi-scale CNN to perform tasks, take images of different scales as input and fusing the results of multiple branches for prediction. In addition, we can also combine the early stop mechanism in neural architecture search (NAS) to find the best partition, and customize a special LViT model for each database, which can greatly increase the computational complexity.

## CONCLUSIONS

In this article, we explore depth palmprint recognition technology. Starting from the most important principal line features of palmprint, WLE data augmentation is proposed to obtain the principal line features of palmprint, resulting good recognition effect, with a maximum gain of 47.88%. At the same time, this article proposes the LViT paradigm for fusion output, which reduces training time and improves recognition accuracy. LViT provides new insights for direction-based methods in the field of palmprint recognition. In addition, this article simulates palmprint collection in the real-world environments, and carries out anti-noise experiments on the noise dataset to verify that LViT maintains high robustness and strong generalization ability in the harsh environment, especially with the

recognition rate on the PolyU II dataset reaching 99.53%, showing minimal impact. Although LViT accelerates model convergence, it also brings inevitable increases in memory pressure and computational cost. In the future, lightweight ViT methods will be explored and binary networks and hash retrieval technology will be introduced to compress the model to improve the retrieval efficiency.

### Funding
The work was supported by the Natural Science Foundation of the Anhui Xinhua University (No. 2023zr003), the Anhui Provincial Quality Engineering Project (No. 2023sx136, No. 2023xsxx356, No. 2020ylzy01) and the Key Research Project of Natural Science in Universities of Anhui Province (No. 2024AH050614, No. 2024AH050621, No. 2024AH050612). There was no additional external funding received for this study. The funders had no role in study design, data collection and analysis, decision to publish, or preparation of the manuscript.

### Grant Disclosures
The following grant information was disclosed by the authors:
Natural Science Foundation of the Anhui Xinhua University: 2023zr003.
Anhui Provincial Quality Engineering Project: 2023sx136, 2023xsxx356, 2020ylzy01.
Key Research Project of Natural Science in Universities of Anhui Province: 2024AH050614, 2024AH050621, 2024AH050612.

### Competing Interests
The authors declare that they have no competing interests.

### Author Contributions
- Hongxia Wang conceived and designed the experiments, performed the experiments, analyzed the data, performed the computation work, prepared figures and/or tables, and approved the final draft.
- Teng Lv performed the computation work, authored or reviewed drafts of the article, and approved the final draft.

### Data Availability
The PolyU II Palmprint Database is available at:

- https://www4.comp.polyu.edu.hk/~csajaykr/palmprint3.htm
- Nebraska-Hewel. (2025). HewelXX/PolyUII: Palmprint PolyU II Dataset (v1.0.0). Zenodo. https://doi.org/10.5281/zenodo.16169889

The IIT Delhi Touchless Palmprint Database is available at: http://www4.comp.polyu.edu.hk/~csajaykr/IITD/Database_Palm.htm

The NTU Contactless Palmprints Database (NTU-CP-v1) is available at: https://github.com/matkowski-voy/Palmprint-Recognition-in-the-Wild

The BJTU_PalmV2 (BJTU-V2) dataset is available at GitHub: https://github.com/HewelXX/Dataset/tree/main/BJTU_V2

XINHUA Palmprint Database is available at GitHub and Zenodo:
- https://github.com/HewelXX/Dataset/tree/main/XINHUA
- Nebraska-Hewel. (2025). HewelXX/Dataset: Palmprint Dataset (v1.0.0) [Data set]. Zenodo. https://doi.org/10.5281/zenodo.15473268

## Supplemental Information

Supplemental information for this article can be found online at http://dx.doi.org/10.7717/peerj-cs.3109#supplemental-information.

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
