# Peer review of "Palmprint recognition based on principal line features"

_PeerJ Computer Science, doi:10.7717/peerj-cs.3109_

## Round 0.1 · original submission · Major Revisions

Please address the reviewer comments in your next revision.

Reviewer 1 ·

Basic reporting

This manuscript proposed a Layer Visual Transformer(LViT) for palmprint recognition. The Wide Line Extraction (WLE) filter is applied to extract the wide palmprint principal lines, and the Gabor filter is used to purify the WLE extraction results. A new database was also established. Experimental analysis shows the effectiveness of the proposed method against various comparison metrics.

Experimental design

1. Some typical examples of different databases are suggested to be added.

2. What is the time complexity of the proposed method? Please analyze it briefly?

3. Please give more details of the experiments, such as the division of each database and the size of the image.

Validity of the findings

-

Additional comments

1. The presentation of this manuscript should be improved significantly.

2. The literature review is incomplete, and more related work from the last two years should be included.

Cite this review as

Reviewer 2 ·

Basic reporting

-

Experimental design

-

Validity of the findings

-

Additional comments

(1) Please check the manuscript carefully to remove the typos, improve the language and format.
E.g. this paper propose -> this paper proposes
...
(2) The abstract is unclear and disordered, so it must be rewritten. “to remove fine lines.” Why should fine lines be removed? “Based on this,” Based on what? Why? “Different blocking strategies are adopted for the input, and all the output results are fused.” What are the blocking strategies? How to fuse?

(3) At the end of the Introduction, Contribution 1, why is it new? What new technologies are used or proposed? The authors should clearly state the contributions and explain why they are contributions/novel.

(4) In Related Work, some paragraphs are too long and difficult to follow, e.g., Para. 2. Please divide them into several short paragraphs to improve readability.

(5) Please provide and label the reference indices (or authors & year) of the compared methods in the figures and tables, so that the readers can judge whether the compared methods are SOTA. The authors

(6) In Table II, the EERs on XINHUA are too high.

Cite this review as

·

Basic reporting

This paper explores deep learning-based palmprint recognition techniques, focusing on the effectiveness of LViT (Lightweight Vision Transformer) and WLE (Weighted Line Extraction) data augmentation. It evaluates different Transformer architectures (ViT, Conformer, PVT-V2, ConvMixer) for palmprint recognition and proposes LViT for better recognition accuracy with fewer training iterations. The research includes multiple experiments, such as baseline tests, comparative analysis, noise resistance evaluation, and computational complexity studies. The results indicate that LViT improves recognition rates while reducing training time, though it comes with increased computational costs.

Experimental design

• Innovative Data Augmentation (the authors use WLE (Wavelet Line Extraction) data augmentation in their experiments). They evaluate its effectiveness on multiple datasets and compare results with and without WLE augmentation. Their findings indicate that WLE improves recognition accuracy, enhances generalization, and strengthens anti-noise performance.
• Improved Model Performance: LViT boosts accuracy while maintaining fast convergence, making it practical for real-time applications.
• Comprehensive Experimentation: The study rigorously tests models in restricted and unrestricted environments, evaluates noise resistance, and compares traditional and deep learning methods.
• Future Potential: The paper suggests lightweight ViTs, binary networks, and hash retrieval to further optimize performance.

Validity of the findings

The paper compares LViT against traditional methods and deep learning-based methods, which is essential for understanding its performance relative to existing techniques.

The use of WLE data augmentation and its effects on different models (ViT, Conformer) provide a well-structured experiment for evaluating the impact of this technique on the accuracy and generalization of palmprint recognition models.

The inclusion of a contrast experiment with four traditional methods and three deep learning methods strengthens the comparative aspect, ensuring that the findings are not biased toward the new method.

The paper demonstrates that LViT performs well on various datasets, including PolyU II, NTU-CP-V1, and BJTU-V2, among others. This shows that the model has strong generalization across different data sources, making the findings more robust and applicable to real-world scenarios.

The anti-noise experiments that simulate environmental disturbances (e.g., rain, dust) validate the model’s ability to handle real-world challenges, suggesting that the model is practical and adaptable.

Additional comments

• A comparison in the form of DET (Detection Error Tradeoff) curves should be presented in your work to better illustrate the trade-offs between the False Rejection Rate (FRR) and the False Acceptance Rate (FAR).
• This method should be applied to 3D palmprint recognition.
• The paper notes that LViT is not the best in all scenarios and emphasizes that the performance is highly dependent on the data used for training. This acknowledgment of the model’s limitations helps contextualize the findings and suggests avenues for further research.
• The potential use of lightweight ViTs and binary networks for reducing computational overhead and improving retrieval efficiency indicates that the authors are considering ways to optimize the model for broader applicability.

Cite this review as

---

## Round 0.2 · Minor Revisions

Please respond to the remaining concern from Reviewer 1.

Reviewer 1 ·

Basic reporting

Thanks for the authors' response for my concerns. One more is that some important related work is still missing, such as cross-domain palmprint recognition.

Experimental design

-

Validity of the findings

-

Additional comments

-

Cite this review as

Reviewer 2 ·

Basic reporting

Good.

Experimental design

Good.

Validity of the findings

Good.

Additional comments

The authors have revised the paper carefully according to the reviewers’ comments. The current version can be accepted now.

Cite this review as

---

## Round 0.3 · accepted · Accept

Dear authors, we are pleased to confirm that you have addressed the valuable feedback from the reviewer to improve your research.

As advised by the reviewer, please consider making the code for LViT and WLE available.

Thank you for considering PeerJ Computer Science and submitting your work.

Kind regards
PCoelho

Reviewer 1 ·

Basic reporting

.

Experimental design

.

Validity of the findings

.

Additional comments

.

Cite this review as

·

Basic reporting

The manuscript presents a significant and well-structured contribution to the field of palmprint recognition using deep learning and line-based features. The proposed WLE (Wide Line Extraction) filter and LViT (Layered Vision Transformer) paradigm are both novel and well-motivated.

Experimental design

The methodology is technically sound and well-structured, with two major innovations:

Wide Line Extraction (WLE) Filter:

A novel technique leveraging directional and width characteristics to emphasize principal lines and suppress fine wrinkles.

Combined with a Gabor filter for post-processing.

Layered Vision Transformer (LViT):

Introduces different patch sizes and fusion methods (sum and max) to improve spatial understanding and convergence.

The paper also includes anti-noise evaluation, a comparative study with traditional and deep learning methods, and an ablation analysis (WLE vs. LViT vs. both). This level of rigor demonstrates a high technical standard.

Validity of the findings

The conclusions presented in the manuscript are well-supported by experimental results. The authors perform extensive evaluation on five publicly available datasets, including their own high-resolution XINHUA dataset. All underlying data and experimental conditions are clearly outlined, with results reproducibly tied to each model and method.

The paper provides detailed comparisons across traditional methods, baseline ViTs, and the proposed LViT.

The authors offer multiple performance metrics (accuracy, EER, ROC, GAR, FAR), which are standard and appropriate for biometric recognition systems.

The dataset links are included in the manuscript, which complies with data availability policies. However, code for LViT and WLE is not yet publicly available, which limits independent verification.

Additional comments

The manuscript presents a significant and well-structured contribution to the field of palmprint recognition using deep learning and line-based features. The proposed WLE (Wide Line Extraction) filter and LViT (Layered Vision Transformer) paradigm are both novel and well-motivated.

Cite this review as